# The Detection and Attribution Model Intercomparison Project (DAMIP v1.0) contribution to CMIP6

Nathan P. Gillett[1], Hideo Shiogama[2], Bernd Funke[3], Gabriele Hegerl[4], Reto Knutti[5], Katja Matthes[6], Benjamin D. Santer[7], Daithi Stone[8], Claudia Tebaldi[9]

[1]Canadian Centre for Climate Modelling and Analysis, Environment and Climate Change Canada, University of Victoria, Victoria, V8W 2Y2, Canada

[2]National Institute for Environmental Studies, Tsukuba, 305-8506, Japan

[3]Instituto de Astrofísica de Andalucía, CSIC, Glorieta de la Astronomía s/n, 18008 Granada, Spain

[4]School of GeoSciences, University of Edinburgh, James Hutton Rd, Edinburgh, EH9 3FE, UK

[5]Institute for Atmospheric and Climate Science, ETH Zürich, Universitätstrasse 16, 8092 Zürich, Switzerland

[6]GEOMAR Helmholtz Centre for Ocean Research Kiel, and Christian-Albrechts-Universität zu Kiel, 24105 Kiel, Germany

[7]Lawrence Livermore National Laboratory, Livermore, California 94550, USA

[8]Lawrence Berkeley National Laboratory, Berkeley, California, 94720, USA

[9]Climate and Global Dynamics Laboratory, National Center for Atmospheric Research, Boulder, CO, 80305, USA

*Correspondence to*: N. P. Gillett (nathan.gillett@canada.ca)

**Abstract.** Detection and attribution (D&A) simulations were important components of CMIP5 and underpinned the climate change detection and attribution assessments of the Fifth Assessment Report of the Intergovernmental Panel on Climate Change. The primary goals of the Detection and Attribution Model Intercomparison Project (DAMIP) are to facilitate improved estimation of the contributions of anthropogenic and natural forcing changes to observed global warming as well as to observed global and regional changes in other climate variables; to contribute to the estimation of how historical emissions have altered and are altering contemporary climate risk; and to facilitate improved observationally-constrained projections of future climate change. D&A studies typically require unforced control simulations and historical simulations including all major anthropogenic and natural forcings. Such simulations will be carried out as part of the DECK and the CMIP6 historical simulation. In addition D&A studies require simulations covering the historical period driven by individual

forcings or subsets of forcings only: such simulations are proposed here. Key novel features of the experimental design presented here include: new historical simulations of aerosols-only, stratospheric-ozone-only, $CO_2$-only, solar-only and volcanic-only forcing, facilitating an improved estimation of the climate response to individual forcing; future single forcing experiments, allowing observationally-constrained projections of future climate change; and an experimental design which allows models with and without coupled atmospheric chemistry to be compared on an equal footing.

## 1 Introduction

Research into the detection and attribution (D&A) of climate change is concerned with identifying forced changes in the observed climate record and assessing the roles of various possible contributors to those observed changes (Hegerl et al. 2010). This research is key for our understanding of anthropogenic climate change, as evidenced by a dedicated chapter in every assessment report of the Intergovernmental Panel on Climate Change (IPCC) since the first report published in 1990. Over this time the resources available to this area of research have developed considerably. Together with longer and improved observations of the climate system, D&A research builds upon the analysis of climate model simulations of the climate in the presence and absence of various factors which are expected to have affected the climate. D&A analyses compare these retrospective predictions against the available observational record, and thus serve as a comprehensive evaluation of our understanding of how the climate system responds to anthropogenic interference. Confidence in our ability to project future climate change hinges strongly on D&A conclusions.

This paper describes a new coordinated international project to conduct D&A simulations with the next generation of climate models, to be conducted as part of the Sixth Coupled Model Intercomparison Project (CMIP6, Eyring et al. 2016). The most basic sets of simulations required for D&A analysis first became available from a large number of climate models through the Third Coupled Model Intercomparison Project (CMIP3, Meehl et al. 2007). At that time, individual forcing simulations allowing attribution investigations were carried out with over half a dozen models (Hegerl et al. 2007; Stouffer et al., 2016), but few of these were publicly available. It was not until the Fifth Coupled Model Intercomparison Project (CMIP5, Taylor et al. 2012) that the full suite of simulations needed for an assessment of the role of anthropogenic forcing, as well as of greenhouse gas emissions specifically, in observed climate trends was conducted using multiple climate models under a common experimental design and, moreover, made publicly available en masse.

D&A studies using CMIP5 simulations underpinned several key high-level findings of the IPCC Fifth Assessment Report (AR5), including for example the assessment that 'it is extremely likely that human influence has been the dominant cause of the observed warming since the mid-20th century', and a figure showing estimated contributions of greenhouse gases, aerosols and natural forcings to observed temperature trends included as one of fourteen summary figures in the summary of the IPCC AR5 Synthesis Report (IPCC, 2014). D&A studies have also underpinned attribution assessments across a range of

variables and regions (Bindoff et al., 2013), and have been used to constrain near-term projected warming (Stott and Jones, 2012; Gillett et al., 2012; Stott et al., 2013; Shiogama et al., 2016), as well as climate system parameters such as Transient Climate Response (TCR) and the Transient Climate Response to Emissions (TCRE) (Allen et al., 2009; Gillett et al., 2013a). Hence D&A results remain of key interest and relevance both scientifically and to policymakers. The only simulations

targeted towards D&A included as part of the CMIP5 experimental design were historical simulations with natural forcing changes only, and historical simulations with greenhouse gas changes only, which were used together with historical simulations and pre-industrial control simulations to support D&A analyses. But research carried out since the design of CMIP5 has highlighted several key research questions which may be addressed by the inclusion of additional simulations in CMIP6.

The separate quantification of greenhouse gas and aerosol contributions to observed global temperature changes is important both for understanding past climate change, and, since aerosol forcing is projected to decline while greenhouse gas forcing increases in the future, for constraining projections of future warming. While some earlier studies were apparently able to clearly separate the influences of greenhouse gases and other anthropogenic forcings on observed temperature changes using individual models (Stott et al., 2006), more recent studies using newer models and a longer period of

observations have identified substantial uncertainties in the separate estimation of greenhouse gas and other anthropogenic contributions, where the other anthropogenic contribution is dominated by aerosols but also includes the response to ozone changes and land use changes in most models (Jones et al., 2013; Gillett et al., 2013a; Ribes and Terray, 2013). These larger uncertainties stem at least in part from uncertainties and inter-model differences in the simulated spatio-temporal pattern of response to aerosols (e.g., Ribes et al. 2015; Boucher et al. 2013), which may have been exacerbated by the large amount of

sampling variability in estimates of the aerosol response derived from a difference between the historical simulations and historical simulations with greenhouse gases only and natural forcings only (e.g., Ribes and Terray, 2013). Simulations of the response to historical changes in aerosols alone will allow the calculation of the response to aerosol forcing with less contamination from internal variability, and without conflating the effects of aerosols with the response to other forcings, most notably ozone and land use changes. Such aerosol-only simulations may be used together with historical simulations

including all forcings and historical simulations with natural forcings only to estimate attributable contributions to observed changes due to natural forcings, due to aerosols, and due to the combined effects of well-mixed greenhouse gases, ozone and land-use changes. Since some part of the greenhouse gas changes is associated with land-use change, and since ozone is a greenhouse gas, grouping these forcing together arguably makes more sense than grouping ozone and land-use change with aerosols. Comparisons of these aerosol-only simulations with proposed Radiative Forcing Model Intercomparison Project

(RFMIP, Pincus et al. 2016) simulations will allow the separation of uncertainty in the simulated aerosol response into a component associated with differences in the simulated distribution of aerosols and a component associated with differences in the simulated climate response to a given aerosol distribution. These advances should allow a more robust quantification of the response to aerosol forcings and its uncertainties.

A further key contribution of D&A studies to the findings of the IPCC AR5 was through the use of observationally-constrained climate projections to inform the assessed range of near-term warming (Stott and Jones, 2012; Gillett et al., 2012; Stott et al., 2013; see also Shiogama, et al. 2016). This is achieved by scaling the projected responses to greenhouse gases and aerosols by their respective regression coefficients derived from a regression analysis over the historical period (Allen et al., 2000; Stott and Kettleborough, 2002; Kettleborough et al., 2007). Such analyses rely on both individual forcing simulations covering the historical period, which were included as part of CMIP5, and individual-forcing simulations of the future, which were not. The latter are required for constraints on projections under scenarios of future emissions in which the relative importance of greenhouse gases to the other anthropogenic forcings changes from that in the historical experiments. Such studies carried out to date have relied on simulations provided by a small number of climate models (Bindoff et al. 2013; Shiogama et al. 2016): the inclusion of future simulations with individual forcings in CMIP6 will allow the derivation of more robust observationally-constrained projections based on a broader range of models, as well as facilitating studies of the aerosol contribution to projected future climate change, which is an area of increasing scientific interest (Shiogama et al., 2010a; Shiogama et al., 2010b; Gillett and von Salzen, 2013; Gagné et al., 2015; Rotstayn et al., 2013; Myhre, et al. 2015).

The experiments proposed as part of DAMIP (Table 1 and Figure 1) will facilitate a number of D&A analyses of anthropogenic and natural forcing influences on historical climate changes. These analyses are expected to address historical changes in temperature, the hydrological cycle, the atmospheric circulation, ocean properties, cryospheric variables, extreme indices and other variables, from global to regional scales (Bindoff et al., 2013). The extension of DAMIP experiments from 2012 in CMIP5 to 2020 with updated climate forcings will support better understanding of the period of reduced warming in the early 21$^{st}$ century (Meehl et al., 2011; Watanabe et al., 2014; Huber and Knutti, 2014; Schmidt et al., 2014) and improve the signal-to-noise ratio for D&A of changes in high-noise variables such as precipitation (Zhang et al., 2007). These DAMIP experiments are particularly relevant to two of the major CMIP6 science questions. Analysis of individual forcing simulations and attribution studies of observed changes will help us to understand how the Earth system responds to forcing. And their use to derive observationally-constrained projections will improve our assessments of future climate change. The DAMIP experiments are also very relevant to the WCRP Grand Challenge on extremes: DAMIP simulations will support attribution studies of changes in temperature, hydrological and other extremes and will improve understanding of the drivers of observed changes in extremes, as well as improving our assessment of the present-day probabilities of extremes. This effort will be further facilitated by using output from DAMIP simulations as input to the C20C+ Detection and Attribution Project (Stone and Pall, 2016) and other similar projects, which use ensembles of simulations of atmosphere-only models driven using observed sea surface temperatures and sea ice, and other similar experiments in which attributable anthropogenic changes are removed from the prescribed SSTs and sea ice to quantify the contribution of anthropogenic changes to individual extreme events. Such studies rely on historical simulations and historical simulations with natural forcings only, such as those included in DAMIP to quantify anthropogenic changes in SSTs. Attribution studies based on

DAMIP output of hydrological changes and cryospheric changes will also address WCRP Grand Challenges on water availability and cryospheric change.

We do not include an analysis plan here, because the field of detection and attribution is well-established, and past experience indicates that individual groups are able to self-organise and to carry out attribution studies on variables and regions of interest. Numerous attribution studies were carried out using the CMIP5 attribution simulations (Bindoff et al., 2013), and it is expected that this number will only increase for DAMIP. D&A activities are coordinated internationally in part by the International Detection and Attribution Group (Barnett et al., 2005), and the long-standing interest of the IPCC in attribution will likely also prompt analysis. Therefore our scope here is restricted to explaining and justifying the planned DAMIP simulations.

## 2 Experimental design

There are two possible frameworks for designing climate model experiments for D&A analysis:  the "only" approach, in which simulations are driven with changes only in the forcing of interest, while all other forcings are held at pre-industrial values; and the "all-but" approach, in which simulations are driven with changes in all forcings except the forcing of interest. An example of the latter is the LUMIP hist-NoLu simulation which includes changes in all forcings but land use change (Lawrence et al., 2016). The two approaches yield equivalent results if additivity holds (i.e. if the climate response to the combined forcing is equal to the sum of the responses to the individual forcings e.g., Rogelj et al. 2012, Knutti and Hegerl 2008) and in the limit of large ensembles, but can differ otherwise. The additivity assumption appears to hold for certain forcings, magnitudes, spatial scales, and variables (e.g., Meehl et al. 2004, Gillett et al., 2004, Shiogama et al. 2013), but may not hold for others (Schaller et al. 2013, Schaller et al. 2014, Marvel et al. 2015b, Knutti and Rugenstein, 2015). We thus recommend that the validity of the additivity assumption is considered in studies using DAMIP simulations. The appropriateness of the "only" or "all-but" approach depends on the scientific question being asked, for instance whether the intention is to understand how the climate system responds to a given factor (in which case the "only" approach would be best), or to detect the contribution of a particular forcing to observed climate change (in which the "all-but" approach may be best). DAMIP will follow a combination of the "only" and "all-but" approaches.  For instance, the response to anthropogenic forcing can be diagnosed from planned DAMIP experiments by taking the difference of the historical and historical natural-only experiments, an "all-but" design.  Whereas the response to greenhouse gas can be determined directly from the historical greenhouse-gas-only experiment, an "only" design. A major consideration is that the proposed design allows direct comparison against the attribution-relevant simulations of CMIP5. The final design reflects a compromise, allowing DAMIP simulations to be used to address a number of scientific questions, whilst limiting the computational demand of the experiments.

## 2.1 Tier 1 Experiments

DAMIP simulations build on the preindustrial control (piControl) simulation which forms part of the DECK, and the CMIP6 Historical Simulation (Eyring et al. 2016) on which all DAMIP historical simulations are based. All simulations used in DAMIP are driven by $CO_2$ concentration rather than $CO_2$ emissions. In common with all other CMIP6 simulations, concentrations of the other well-mixed greenhouse gases (WMGHGs) are also specified in DAMIP simulations. We request at least 3 ensemble members with different initial conditions for each historical individual forcing experiment, and recommend that modeling groups which cannot afford to do this for all requested runs start by carrying out at least 3-member ensembles of the Tier 1 simulations. We recommend that ensemble members are initiated by choosing well-separated initial states from a control simulation.

In order to maximise the signal-to-noise ratio, and to facilitate the comparison with the most recent climate observations, it is often desirable to include data from the most recent years in D&A analyses. Given that the CMIP6 historical simulation finishes in 2014, we therefore request that modelling groups extend all DAMIP historical simulations to 2020 using the SSP2-4.5 forcings. A similar approach was applied in CMIP5 with D&A simulations extended from 2006 to 2012 using RCP 4.5 forcings. While this approach has the disadvantage that forcings in the final years of the simulations are estimated rather than directly observed, and in the case of CMIP5 there was discussion about whether differences in forcing over the post-2005 period could contribute to differences in simulated and observed temperature trends over the early 21$^{st}$ century (Santer et al., 2013; Huber and Knutti, 2014; Schmidt et al., 2014), in practice, barring a major volcanic eruption such forcings are unlikely to diverge strongly from reality (see Box 9.2 of Flato et al. 2013). Moreover, as long as the use of projected forcings in the post-2014 period is made clear, as it is here, investigators can make their own informed decision of whether or not to include the post-2014 period in their analyses. SSP2-4.5 was chosen because of its intermediate level of greenhouse gas emissions and its future aerosol and land-use changes which are more representative of a broader range of SSP-based integrated assessment model projections. A finish date of 2020 was chosen because it will allow contemporary observations to be included in detection and attribution studies cited in the Sixth IPCC Assessment Report, without extending too far into the future. Four experiments are requested under Tier One, each of which should cover the 1850-2020 period and comprise at least three simulations with different initial conditions.

**CMIP6 historical (1850-2014) & SSP2-4.5 of ScenarioMIP (2015-2020) (historical)**:  This includes the CMIP6 historical simulations (Eyring et al. 2016), the extension of those simulations to 2020 under the SSP2-4.5 scenario, and the generation of at least two additional (thus three total) members with different initial conditions.  For brevity, we call this CMIP6 historical + SSP2-4.5 experiment "historical" in this paper. Modelling groups should provide the output data under the labels of the CMIP6 historical runs (1850-2014) and the SSP2-4.5 runs of ScenarioMIP (O'Neill et al., 2016) (2015-2020). Time-

evolving solar forcing, and stratospheric aerosol ramping up towards the piControl background level should be prescribed over the 2015-2020 period as specified by ScenarioMIP (O'Neill et al., 2016).

**hist-nat**: These historical natural-only simulations resemble the historical simulations but are forced with only solar and volcanic forcings from the historical simulations, similarly to the CMIP5 historicalNat experiment. Together with the historical and piControl simulations, such simulations will allow the attribution of observed changes to anthropogenic and natural influences.

**hist-GHG**: These historical greenhouse-gas-only simulations resemble the historical simulations but instead are forced by *well-mixed* greenhouse gas changes only, similarly to the CMIP5 historicalGHG experiment. historical, hist-nat and hist-GHG will allow the attribution of observed climate change to natural, greenhouse gas and other anthropogenic components. Models with interactive chemistry schemes should either turn off the chemistry or use a preindustrial climatology of stratospheric and tropospheric ozone in their radiation schemes. This will ensure that tropospheric and stratospheric ozone are held fixed in all these simulations, and simulated responses in models with and without coupled chemistry are comparable. By comparison, in CMIP5 some models included changes in ozone in their greenhouse-gas only simulations while others did not (Gillett et al., 2013; Jones et al., 2013; Bindoff et al., 2013), making it harder to compare responses between models.

**hist-aer**: These historical aerosol-only simulations resemble the historical simulations but are forced by changes in aerosol forcing only. As discussed in the introduction, such simulations should allow the response to aerosols to be better constrained, and physically understood, and may also allow the response to greenhouse gases to be better constrained (e.g., Ribes et al., 2015). Two experimental designs are proposed for anthropogenic aerosol-only runs depending on whether the model concerned includes a complete representation of atmospheric chemistry. For models in which greenhouse gas concentrations do not influence aerosol concentrations, and aerosol precursor concentrations do not influence ozone concentrations, we request historical simulations forced by anthropogenic aerosol concentrations only or aerosol and aerosol precursor emissions only as in the historical simulation (sulfur dioxide, sulfate, black carbon (BC), organic carbon (OC), ammonia, carbon monoxide, NOx and non-methane volatile organic compounds (NMVOCs)). For models with interactive atmospheric chemistry in which aerosol and greenhouse-gas concentrations interact, we recommend an alternative experimental design. Changes in well-mixed GHGs, aerosol precursors and ozone precursors should be prescribed as in the historical simulations. However, in the radiation scheme, the concentrations of well-mixed-GHGs and the ozone climatology from the piControl runs should be used. This procedure will allow the simulation of aerosol burdens and the associated climate influence consistent with the historical simulations, hence allowing output from historical simulations and aerosol-only simulations to be meaningfully compared.

**2.2 Tier 2 Experiments**

DAMIP Tier 2 Experiments include experiments to enable observationally-constrained projections (Stott and Kettleborough, 2002; Stott and Jones, 2012; Gillett, et al. 2012; Shiogama et al., 2016) and experiments to facilitate the attribution of observed changes to stratospheric ozone changes (Gillett et al., 2013; Lott et al., 2013). Before performing the future simulations of DAMIP, modelling groups are asked to complete at least one SSP2-4.5 simulation of ScenarioMIP up to 2100. Minimum ensemble sizes are three for the historical simulations and one for the future simulations, though modelling groups are encouraged to run larger ensembles if resources allow.

**ssp245-GHG**:  This comprises an extension of at least one hist-GHG simulation to 2100 using the SSP2-4.5 well-mixed greenhouse gas concentrations.  ssp245-GHG, ssp245-nat (see Section 2.3) and SSP2-4.5 will allow the simulated future responses to greenhouse gases, natural-forcing and other forcing to be separated and constrained separately based on regression coefficients derived from observations, hence allowing observationally-constrained projections to be derived. As in hist-GHG, models with interactive chemistry schemes should either run with the chemistry scheme turned off or use a preindustrial climatology of ozone in the radiation scheme.

**hist-stratO3**:  These simulations resemble the historical simulations but are forced by changes in stratospheric ozone concentrations only. They will allow an improved characterization of the response to stratospheric ozone changes, which have played an important role in driving circulation changes in the Southern Hemisphere and temperature changes in the stratosphere, as well as facilitating attribution studies of the response to stratospheric ozone change (e.g., Gillett et al., 2013b). Such experiments were not included as standard in CMIP5, although a small number of modelling groups carried them out. In models with coupled chemistry, the chemistry scheme should be turned off, and the simulated ensemble mean monthly mean 3D stratospheric ozone concentrations from the historical simulations should be prescribed, since previous studies have indicated that the 3-D structure of ozone trends is an important driver of the tropospheric response (e.g. Waugh et al., 2009, Crook et al., 2008). Tropospheric ozone should be fixed at 3D long-term monthly mean piControl values, with grid cells having an ozone concentration below 100 ppbv in the piControl climatology for a given month classed as tropospheric. In models without coupled chemistry the same stratospheric ozone prescribed in the historical simulation should be prescribed. Note that DAMIP does not include simulations isolating the effects of tropospheric ozone changes.

**ssp245-stratO3**:  These simulations are extensions of the hist-stratO3 simulations to 2100 following the ozone concentrations specified for the SSP2-4.5 scenario. Stratospheric ozone is projected to recover following the successful implementation of the Montreal Protocol and its amendments (WMO, 2014). These simulations will facilitate a robust multi-model assessment of the climate effects of this recovery on Southern Hemisphere climate and stratospheric temperature.

### 2.3 Tier 3 Experiments

DAMIP Tier 3 experiments consist of solar, volcanic and $CO_2$ individual forcing experiments, an extension of the aerosol-only and natural-only simulation to 2100 and perturbed forcing experiments. Minimum ensemble sizes are three for the historical simulations and one for the future simulations.

**hist-sol:** These simulations resemble the hist-nat simulations except that hist-sol simulations are driven by solar forcing only. The potential importance of solar forcing in particular for regional climate variability is becoming increasingly evident (Gray et al., 2010; Seppälä et al., 2014). Because of its prominent approximately 11-year cycle, solar variability could offer a degree of predictability for regional climate variability. Foreseeable fluctuations in solar output could help reduce the uncertainty of future regional climate predictions on decadal time scales. However there are still large uncertainties in the

atmospheric solar signal and its transfer mechanisms including changes in total and spectral solar irradiance as well as in solar-driven energetic particles, and there is uncertainty in the significance and amplitude of the climate response to a solar changes due to the presence of internal variability. Recent work suggests a lagged response in the North Atlantic European region due to atmosphere-ocean coupling (Gray et al., 2013, Scaife et al., 2013) as well as a synchronization of decadal NAO variability through the solar cycle (Thiéblemont et al., 2015). Recent modeling efforts have made progress in defining the

pre-requisites to simulate solar influence on regional climate more realistically but the lessons learned from CMIP5 show that a more systematic analysis of climate models within CMIP6 is required to better understand the differences in model responses to solar forcing (Mitchell et al., 2015; Misios et al., 2016; Hood et al., 2015). In particular the role of solar induced ozone changes and the need to prescribe spectrally resolved solar irradiance variations and therefore the need for a suitable resolution in the model's shortwave radiation scheme is becoming increasingly evident. The proposed hist-sol experiment

will facilitate the characterization of each model's solar signal and allow for a more systematic analysis of the differences in model responses. The hist-sol simulations will also allow the estimation of the effect of solar forcing on observations separately from that of volcanism (rather than in combination with volcanism using hist-nat), which is essential for the quantification of the solar signal over the historical period.

**hist-volc**: The hist-volc simulations resemble the hist-nat simulations except that the hist-volc simulations are driven by stratospheric aerosol forcing only. The hist-volc experiments will facilitate detection and attribution studies on volcanic influence. Careful evaluation of a model's volcanic response may inform its use for geoengineering simulations, such as those in GeoMIP (Kravitz et al., 2013). In addition, it will be possible to test additivity of the responses to these natural forcings (Gillett et al., 2004; Shiogama et al., 2013) by comparing the sum of the hist-volc and hist-sol responses with the

hist-nat response.

**hist-CO2:** These are historical simulations driven by observed changes in $CO_2$ concentration only as in historical. One approach to observationally-constraining the ratio of warming to cumulative $CO_2$ emissions, a policy-relevant climate metric known as the Transient Climate Response to Emissions (TCRE), requires an estimate of historical $CO_2$-attributable warming,

but detection and attribution analyses typically only provide an estimate of warming attributable to changes in all well-mixed greenhouse gases (Gillett et al., 2013a). Together with hist-GHG simulations these simulations would allow the ratio of $CO_2$-attributable to GHG-attributable warming to be estimated, and hence more robust estimates of TCRE to be obtained. Further, observationally-constrained estimates of the Transient Climate Response (TCR) typically assume a perfect correlation between TCR and warming in hist-GHG across models, but in fact there is considerable spread in this ratio (Gillett et al., 2013a). These simulations will allow the reasons for this spread to be investigated, and hence for the better characterisation of uncertainties in TCR.

**ssp245-aer**: This involves an extension of at least one of the hist-aer simulations to 2100 using SSP2-4.5 aerosol concentrations/emissions. Together with SSP2-4.5 and ssp245-nat, this will allow observationally-constrained projections of future climate to be derived based on separating past and future climate change into components associated with natural forcings, aerosols, and other anthropogenic forcings (well-mixed greenhouse gases, ozone and land-use change). Such an approach may give more accurate estimates of uncertainties than the more usual approach in which climate change is separated into natural, well-mixed greenhouse gas, and other anthropogenic (aerosols, ozone and land-use change) components (e.g. Ribes et al., 2015). These simulations will also allow the more robust characterisation of the response to future aerosol changes, without conflating these changes with the responses to ozone and land-use changes.

**ssp245-nat**: This involves an extension of at least one of the hist-nat simulations to 2100 following SSP2-4.5 solar and volcanic forcing. The future solar forcing data recommended for CMIP6 has a downward trend (Matthes et al., 2016), and stratospheric aerosol is prescribed to ramp up from its level in 2014 to the background level specified in piControl over the 2015-2025 period (O'Neill et al., 2016). ssp245-nat may be used to investigate effects of these forcing changes on future climate change projections. Together with SSP2-4.5 and ssp245-GHG, ssp245-nat will allow observationally-constrained projections of future climate to be derived based on separating past and future climate change into components associated with natural forcing, well-mixed greenhouse gases and other anthropogenic forcing factors.

**hist-all-aer2 and hist-all-nat2**: The final two sets of simulations, hist-all-aer2 and hist-all-nat2, are identical to the historical simulation (including the extension to 2020 with SSP2-4.5), except that they contain alternative estimates of the aerosol forcing and natural forcings, respectively. Standard attribution analyses sample over internal variability, and in some cases sample over model uncertainty and observational uncertainty (e.g., Bindoff et al., 2013). However, if these analyses use a multi-model ensemble in which all models use the same set of forcings, then the contribution of forcing uncertainty to the full uncertainty in the results is neglected. Hence we propose simulations with different estimates of historical forcings to explore this source of uncertainty. We focus here on the uncertainties in aerosol emissions/concentrations and natural forcings since these sources of forcing uncertainty are expected to be the most important for global climate. Investigators could for example carry out attribution analyses using the hist-all-aer2, hist-GHG and hist-nat simulations to address the

contribution of aerosol forcing uncertainty to global attribution results, and similarly use the hist-all-nat2, hist-GHG and hist-aer simulations to address the contribution of natural forcing uncertainty to global attribution results. These simulations could also be used to examine the role of forcing uncertainties in simulating climate trends over particular periods, such as global warming over the early 21[st] century. The exact method for sampling uncertainty in the tropospheric aerosol, volcanic, and solar forcings is being developed in cooperation with the groups developing forcing data sets for CMIP6.

## 3. Synergies with other MIPs

Synergies between the CMIP6 endorsed MIPs are important for maximising the value of CMIP6 and allowing us to address the science questions of the WCRP grand challenges. Table 2 shows potential synergies between DAMIP, the other MIPs and other relevant research activities.

The Decadal Climate Prediction Project (DCPP, Boer et al. 2016) and DAMIP together propose the enlargement of the ensemble size of historical and SSP2-4.5 to investigate the importance of internal variability in the past and the near future, providing reduced total computing costs if both MIPs are pursued. DCPP and the Global Monsoons Modeling Intercomparison Project (GMMIP, Zhou et al. 2016) have proposed the pacemaker 20th century historical runs (Kosaka and Xie, 2013) to understand the influences of the Interdecadal Pacific Oscillation (IPO) and the Atlantic Multidecadal Oscillation (AMO) on historical climate changes. The combination of these pacemaker experiments and DAMIP experiments will facilitate assessments of the relative contributions of external forcing factors and internal variability to historical climate change.

Closely collaborating with DAMIP, the Effective Radiative Forcing subproject of RFMIP (RFMIP-ERF, Pincus et al. 2016) has proposed experiments to estimate effective radiative forcing for all forcings combined, well-mixed greenhouse gas forcing, natural forcing and anthropogenic aerosol forcing. Combining radiative forcing estimated from RFMIP-ERF and transient climate responses from DAMIP, we can investigate transient climate sensitivities (or forcing efficacies) (Hansen et al., 2005; Yoshimori and Broccoli, 2008; Shindell, 2014) for those forcing agents, which may be used to help derive observational constraints on TCR and effective climate sensitivity (Shindell, 2014; Kummer and Dessler, 2014; Gregory et al. 2015; Marvel et al., 2015a).

LUMIP (Lawrence et al. 2016) proposes historical all forcing experiments without land-use land-cover changes. Combinations of these and the DAMIP experiments allow the separation of climate responses to land-use land-cover changes and the other forcing agents. Lastly, while VolMIP (Zanchettin et al. 2016) includes simulations of individual eruptions it does not include simulations of the transient response to historical eruptions, except for the Tambora period in the 19th century. hist-volc of DAMIP allows better validation of long-term transient effects against observations,

## 4. New variables requested by DAMIP

The specific questions addressed by the hist-sol experiment, in particular the attribution of differences in the model responses to solar forcing and its link to different transfer mechanisms, require additional model output related to the radiation scheme and calculated or prescribed ozone chemistry. This includes zonal mean short- and longwave heating rates, as well as ozone fields (prescribed or interactively calculated). Further, a reduced set of new chemistry variables has been

proposed for models with interactive chemistry schemes, including $O_2$ and $O_3$ photolysis rates, as well as odd oxygen total loss and production rates. These new variables may also be of interest for investigating and understanding the response to other forcings in the DAMIP simulations.

## 5. Summary

DAMIP will coordinate the climate model simulations needed to more robustly attribute global and regional climate change to anthropogenic and natural causes, to derive observationally-constrained projections of future climate change, and to improve understanding of the mechanisms by which particular forcings affect climate. The Tier One simulations differ from those included in CMIP5 only by the inclusion of a set of simulations with aerosol changes only, which will help constrain the climate response to aerosol forcing, and also by an experimental design that ensures that results from models with and

without coupled chemistry are comparable. Tiers Two and Three include individual forcing simulations covering the future through to the end of the century, which are needed to observationally constrain the future response to greenhouse gases, aerosols, stratospheric ozone and natural forcing based on observed historical changes. These tiers also include additional simulations of the response to historical variations in stratospheric ozone, $CO_2$, volcanoes and solar forcing individually, and perturbed forcing simulations to allow the contribution of forcing uncertainty to uncertainty in attribution results to be

assessed for the first time.

**DAMIP Website:** Updated details on the project and its progress will be available at http://damip.lbl.gov.

**Data Availability:** The model output from the DAMIP simulations described in this paper will be distributed through the

Earth System Grid Federation (ESGF) with digital object identifiers (DOIs) assigned. As in CMIP5, the model output will be freely accessible through data portals after registration. In order to document DAMIP's scientific impact and enable ongoing support of DAMIP, users are asked to acknowledge CMIP6, DAMIP, the participating modelling groups, and the ESGF centres (see details on the CMIP Panel website at http://www.wcrp-climate.org/index.php/wgcm-cmip/about-cmip).

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

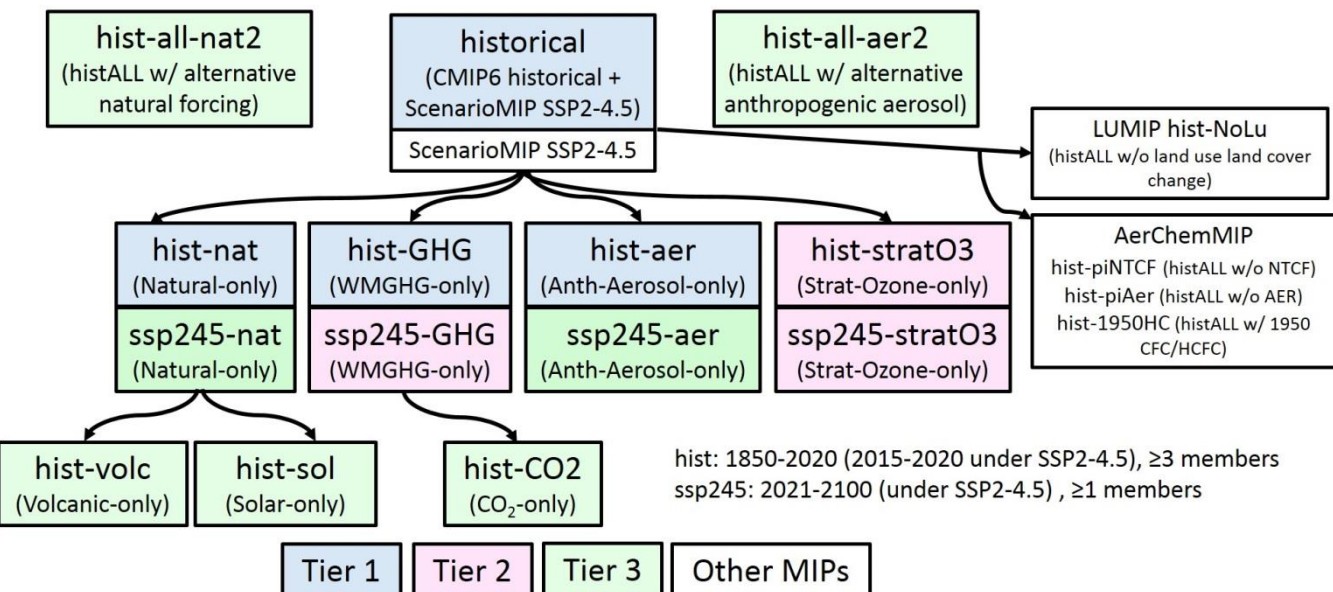

**Figure 1.** Schematic of the relationships of the various experiments proposed under DAMIP and other MIPs. Solid arrows indicate the decomposition into separated forced responses.

**Table 1.** List of proposed DAMIP experiments

| Name of exp. | Description [Forcing agents perturbed] | Tier | Start year | End year* | Min. Ens. size |
|---|---|---|---|---|---|
| CMIP6 historical simulation and SSP2-4.5 | Enlarging ensemble size of the CMIP6 historical simulations (1850-2014) and the SSP2-4.5 simulations of ScenarioMIP (2015-2020). While these simulations are called historical in this paper for readability, please provide the output data separately as the CMIP6 historical simulation (-2014) and SSP2-4.5 (2015-). [As historical: WMGHGs, BC, OC, $SO_2$, $SO_4$, NOx, $NH_3$, CO, NMVOC, nitrogen deposition, ozone, stratospheric aerosols, solar irradiance, land use] | 1 | 1850 | 2020 | 3 |
| hist-nat | Natural-only historical simulations [Solar irradiance, stratospheric aerosol] | 1 | 1850 | 2020 | 3 |
| hist-GHG | Well-mixed greenhouse-gas-only historical simulations [WMGHGs] | 1 | 1850 | 2020 | 3 |
| hist-aer | Anthropogenic-aerosol-only historical simulations [BC, OC, $SO_2$, $SO_4$, NOx, $NH_3$, CO, NMVOC] | 1 | 1850 | 2020 | 3 |
| ssp245-GHG | Extension of at least one histGHG simulation through the 21st century using the SSP2-4.5 greenhouse gas concentrations [WMGHGs] | 2 | 2021 | 2100 | 1 |
| hist-stratO3 | Stratospheric-ozone-only historical simulations [Stratospheric ozone] | 2 | 1850 | 2020 | 3 |
| ssp245-stratO3 | Extension of at least one hist-stratO3 simulation through the 21st century using SSP2-4.5 stratospheric ozone concentrations [Stratospheric ozone] | 2 | 2021 | 2100 | 1 |
| hist-sol | Solar-only historical simulations [Solar irradiance] | 3 | 1850 | 2020 | 3 |
| hist-volc | Volcanic-only historical simulations [Stratospheric aerosol] | 3 | 1850 | 2020 | 3 |
| hist-CO2 | $CO_2$-only historical simulations [$CO_2$] | 3 | 1850 | 2020 | 3 |
| ssp245-aer | Extension of at least one hist-aer simulation through the 21st century using SSP2-4.5 tropospheric aerosol concentrations/emissons [BC, OC, $SO_2$, $SO_4$, NOx, $NH_3$, CO, NMVOC] | 3 | 2021 | 2100 | 1 |
| ssp245-nat | Extension of at least one hist-nat simulation through the 21st century using SSP2-4.5 solar and volcanic forcing [Solar irradiance, stratospheric aerosol] | 3 | 2021 | 2100 | 1 |
| hist-all-aer2 | historical with alternate estimates of anthropogenic aerosol emissions/concentrations [As historical: WMGHGs, BC, OC, $SO_2$, $SO_4$, NOx, $NH_3$, CO, NMVOC, nitrogen deposition, ozone, stratospheric aerosols, solar irradiance, land use] | 3 | 1850 | 2020 | 3 |
| hist-all-nat2 | historical with alternate estimates of solar and volcanic forcing [As historical: WMGHGs, BC, OC, $SO_2$, $SO_4$, NOx, $NH_3$, CO, NMVOC, nitrogen deposition, ozone, stratospheric aerosols, solar irradiance, land use] | 3 | 1850 | 2020 | 3 |

*2015-2100 segments of the simulations are driven by the SSP2-4.5 emission scenario.

**Table 2.** Synergies with DECK, CMIP6 historical and other MIPs

| MIP or Project | Simulations in MIP | Simulations in DAMIP | Area of synergy |
|---|---|---|---|
| DECK[1] | piControl, 1pctCO2 | All | piControl is essential for estimating internal variability. Thus we recommend that modelling groups perform a 500-year or longer piControl run to allow robust estimates of internal variability. 1pctCO2 is needed for observationally-constrained estimates of TCR and TCRE. |
| CMIP6 historical simulations[1] | All | historical, hist-nat, hist-GHG, hist-aer, hist-stratO3, hist-volc, hist-sol, hist-CO2 | All the historical experiments of DAMIP are based on the CMIP6 historical simulations and allow refined understanding of the contribution of individual forcing components to climate variations and change in the CMIP6 historical simulations. |
| ScenarioMIP[2] | SSP2-4.5 | historical, hist-nat, hist-GHG, hist-aer, ssp245-GHG, ssp245-aer, ssp245-nat | All DAMIP scenario simulations are based on SSP2-4.5, and together with SSP2-4.5 they allow observationally-constrained future projections to be derived. |
| AerChemMIP[3] | hist-piNTCF hist-piAer hist-1950HC | historical, hist-nat | hist-piNTCF (AerChemMIP) is the same as historical (DAMIP) but with 1850 aerosol and ozone precursors. In hist-piAer, only the aerosol precursors are kept at 1850, while the ozone precursors follow the historical. hist-1950HC is the same as historical but with 1950 CFC and HCFC concentrations.. These AerChemMIP simulations may be used with DAMIP simulations to attribute observed changes to changes in emissions of aerosol precursors, ozone precursors, or CFC and HCFCs, in combination with natural forcings and other anthropogenic forcings. |
| DCPP[4] | Historical+SSP2-4.5 C1.9 (Pacemaker Pacific experiment) C1.10 (Pacemaker Atlantic experiment) | historical | DCPP proposes a 10-member ensemble of historical up to 2030 also extended with SSP2-4.5. The combinations of DAMIP and DCPP/GMMIP experiments allow the assessment of the relative contributions of external forcing factors and the response to the PDO and AMO to historical climate change. |
| GMMIP[5] | AMIP20C, HIST-IPO, HIST-AMO | historical, hist-nat, hist-GHG, hist-aer | The combinations of DAMIP and DCPP/GMMIP allow the assessment of the relative contributions of external forcing factors and internal variability to historical climate change. |
| LUMIP[6] | hist-NoLu | historical | hist-NoLu) of LUMIP and historical will allow the separation of the effects of land-use changes and the response to other forcings. |
| RFMIP-ERF[7] | RFMIP-ERF-HistAll, RFMIP-ERF-HistNat, RFMIP-ERF-HistGHG, RFMIP-ERF-HistAer | historical, hist-nat, hist-GHG, hist-aer, ssp245-GHG, ssp245-aer, ssp245-nat | Combining radiative forcing estimated from RFMIP-ERF and transient climate responses from DAMIP, we can investigate how feedbacks and adjustments vary with forcing factors. |
| RFMIP-SpAer[7,8] | RFMIP-SpAerO3-all, RFMIP-SpAerO3-aer | historical, hist-nat, hist-aer | Combinations of DAMIP and RFMIP-SpAer will allow us to separate uncertainties in climate response based on specified aerosol evolution from the overall uncertainties in climate response to specified aerosol precursor emissions. |
| GeoMIP[9] & VolMIP[10] | All | hist-volc | The volcanic response of models can be validated against observations using histVLC, whereas GeoMIP experiments cannot. Thus histVLC experiments will provide useful context for interpreting simulated responses to stratospheric aerosol across models in the GeoMIP experiment. While VolMIP includes simulations of individual eruptions, it does not include simulations of the transient response to historical eruptions and its focus is on 19th century eruptions. histVLC facilitates validation of long-term transient effects against observations. |

The reference papers are [1]Eyring et al. (2016), [2]O'Neill et al. (2016), [3]Collins et al. (2016), [4]Boer et al. (2016), [5]Zhou et al. (2016), [6]Lawrence et al. (2016), [7]Pincus et al. (2016), [8]Stevens et al. (2016), [9]Kravitz et al. (2015) and [10]Zanchettin et al. (2016)