# Peer review of "The Detection and Attribution Model Intercomparison Project (DAMIP v1.0) contribution to CMIP6"

_Geoscientific Model Development, 2016_

## Short Comment (SC1) · 14 Apr 2016

Dear authors,

In agreement with the CMIP6 panel members, the Executive editors of GMD would like to establish a common naming convention for the titles of the CMIP6 experiment description papers.

The title of CMIP6 papers should include both the acronym of the MIP, and CMIP6, so that it is clear this is a CMIP6-Endorsed MIP.

Additionally, we strongly recommend to add a version number to the MIP description. The reason for the version numbers is so that the MIP protocol can be updated later, normally in a second short paper outlining the changes. See, for example:

[Figure]

http://www.geosci-model-dev.net/special_issue11.html,

Good formats for the title include:

'XYZMIP (v1.0) contribution to CMIP6: Name of project'

or

'Name of Project (XYZMIP v1.0) contribution to CMIP6'

If you want to include a more descriptive title, the format could be along the lines of,

'XYZMIP (v1.0) contribution to CMIP6: Name of project - descriptive title'

or

'Name of Project (XYZMIP v1.0) contribution to CMIP6: descriptive title.'

When you revise your manuscript, please correct the title of your manuscript accordingly.

Yours,

Astrid Kerkweg

---

## Referee Comment (RC1) · A. Ribes (Referee) · 12 May 2016

Review of Manuscript by N.P. Gillett and colleagues "Detection and Attribution Model Intercomparison Project (DAMIP)"

The manuscript provides a clear and comprehensive presentation of the design of the CMIP6-DAMIP exercise, i.e. the numerical experiment devoted to the Detection and Attribution questions within CMIP6. The main motivations underpinning Detection and Attribution and their historical background are nicely described, as well as the main features of previous Model Intercomparison exercises. The new design that is proposed here introduces new types of experiments that will enable or facilitate new analyses of both historical and future changes. In my view, historical AER-only, SOZ-only, and CO2-only experiments, and the extension of individual forcing simulation to the 21st

century are very attractive novelties. Most importantly, the manuscript provides a very accurate description of each experiment, with detailed information on many technical aspects that are needed to realise those experiments. Overall, this is a very useful contribution for the Detection and Attribution community, and the modelling groups that will take part to this MIP.

I have several specific comments, which can all be considered as minor comments which the authors are free to take into consideration or not, although two of them are more substantial.

Substantial comments

1. About event attribution. The paper provides a detailed enumeration of scientific questions for which DA experiments have been used in the past. Noticeably, the attribution of single weather / climate events has not been mentioned. I don't know if this was intentional or not. This area of event attribution has received much attention recently - maybe even excessively, I agree. What I would call the "dominant" method used to perform event attribution calculations involves large ensembles of forced atmospheric experiments - which are not DAMIP style simulations. But the counter-factual (ie NAT-only) SST are usually constructed using a common D&A analysis of long-term changes, in an ANT vs NAT decomposition. DAMIP like experiments are required here. Additionally, there have been efforts to assess how the final results (eg FARs) depend on the assumed (ANT and NAT) response patterns, which basically requires a MIP with histNAT simulations from different models. Overall, I think this is an important application, which requires DAMIP, and which could be mentioned.

2. Comprehensive list of external forcings and how they are clustered into subsets. The paper provides a very clear list of experiments, and in most cases, a clear list of forcing agents to be considered in these experiments. However, I suggest it might be useful to provide an "as exhaustive as possible" list of external forcings, and the individual forcing experiments in which they are supposed to be included (eg which are

classed as "Aerosols", etc). I think this might be useful for several reasons: - there might be some inconsistencies with AR5, e.g., with respect to what is called aerosols. According to, e.g., the AR5 Fig 8.17, aerosols do not include NOx or NMVOC, while these species are included in the histAER experiment that is proposed here. The status of tropospheric Ozone with respect to GHGs was also somewhat unclear in CMIP5. - some external forcings are not included in any DAMIP individual forcing simulations. This applies to LU, which is being treated in LUMIP, but also to Tropospheric Ozone, and maybe other short lived gases like CO. An exhaustive list could make this clearer. - NetCDF files from CMIP5 were usually specifying a list of forcing agents explicitly (at least for GHG species, in my memory), so this work would have to be done at some point anyway.

Other minor comments

* p3 l15 and l17: I suspect that the appropriate reference is Ribes and Terray 2013, instead of Ribes et al. 2015 * p3 l21-22 "with those of ozone and land use changes": I suggest replacing by "with the response to other external forcings, most notably ozone and land use" or something of that effect, as I'm not sure that all forcings can be put into the categories GHG, AER, NAT, OZ and LU (see also comment 2). * p4 l2: "the other" is written twice * p5 1st paragraph: I suggest adding somewhere something like "The two approaches ["only" vs "all-but"] are equivalent if additivity holds, but might differ otherwise." * p5 Sentence l5-7: it is not clear to me that the "all-but" approach is more appropriate than the "only" one for the second question mentioned... Is it what the authors want to say? As an alternative, causality theory wight be mentioned explicitly, as it seems to be the main motivation leaning towards an "all-but" approach. * p5 l14: "linear additivity" has been indeed commonly discussed in the literature but it seems to me that, strictly speaking, only "additivity" is assumed in the experimental design. "Linearity", in my view, is more related to the use of analysis techniques based on linear regression, like optimal fingerprinting. * p5 l21-22: Does this also apply to GHGs other than CO2? * p6 histALL: This is probably well specified in other CMIP6

documents, but I think it might be useful to add a quick description of the NAT forcing recommended in SSP. To my knowledge, there were no clear recommendations in CMIP5 regarding the volcanic forcing. Additionally, at least one modelling center decided to run historicalExt experiments with no volcanoes (consistent with the observation of no major eruption when the run was realised in 2011/12), while the RCPs were run with a volcanic background. This led to historicalExt runs which differ from the corresponding RCP runs over their common period. My understanding of the description given here is that such a discrepancy should be avoided in CMIP6, and I think it would be useful to state this even more clearly. * p6 histGHG: To make the point even clearer, I suggest adding a sentence such as "Ozone (tropospheric and stratospheric) is excluded from GHG species [and is therefore supposed to remain roughly constant in these experiments]", eg at l21. * p7 l17: maybe add "(see Tier 3)" after ssp245NAT * Tiers 2 and 3: information on minimum ensemble sizes seems less precise for those Tiers if compared to Tier 1 - maybe it might be said somewhere that the general rule is at least 3 for historical, at least (only) 1 for SSP... Or maybe I missed it? * Lastly, I also suggest adding somewhere something like "Tropospheric ozone (and possibly other external forcings, if any) is not considered in any of the historical experiments driven by subsets of forcings which are proposed in DAMIP.". Consequently, quantifying the response to this forcing would require subtraction, with a possible confounding effect related to non-additive responses to other forcings. Note that this choice doesn't seem inappropriate to me, but just it would be useful to make it clear in order to prevent misinterpretation.

Hope this helps.

---

## Referee Comment (RC2) · Anonymous Referee #2 · 13 May 2016

**Summary**

The authors describe the Detection and attribution model intercomparison project (DAMIP) that is planned to contribute to the Climate Model Intercomparison Project phase 6 (CMIP6). The overall aim of DAMIP is explained as a framework for climate research institutions to use to produce a variety of climate simulations of the past 100 or so years, following a consistent experimental design. The authors propose that the experiments will aid in the understanding and attribution of observed past changes, and help in the constraining of projections of future climate change. A number of different experiments are devised, with different priorities, which build on those produced for CMIP5.

[Figure]

This plan is very much welcome. Having a clear design will hopefully build on the qualified success of CMIP5. What is proposed is very clearly presented.

The manuscript is generally clearly written, with only a few typographic corrections needed - see technical comments below.

I recommend acceptance, after some relatively minor revisions - clarifications about the design and motivation for the choice of experiments.

I do have concerns about some of the actual proposed experiments, but I suspect the experimental design is set in stone, and the reviewer opinions and the other open comments may not be able to influence what experiments are included in the plan. However, I strongly urge the authors to consider the below comments about the specifics of the design and consider if tweaks can be made to improve the efficacy of what is planned and increase institutional engagement.

**General comments**

I fear, with respect to resources required, that the proposal is somewhat over ambitious. Just for Tier 1 the number of experiments and initial condition ensembles required, >1870 model years, may substantially limit the number of models participating in DAMIP. ScenarioMIP (O'Neill GMDD [2016]) say "the success of ScenarioMIP lies in the broad participation of the CMIP6 modelling groups in Tier 1 experiments...". Should the DAMIP plan have the same ambition? A much smaller population of experiments/ensembles for a 'Tier 0', to focus on a few important scientific questions, could encourage as wide a range of models to take part in DAMIP as possible. The remaining tiers can then be populated by institutions with more resources. It would be a shame

to miss the opportunity to design experiments that would encourage greater institution involvement than there was for CMIP5 detection analyses. A 'lessons learned' exercise and finding out why some institutions didn't produce 'detection' experiments for CMIP5 might have been helpful.

There are several experiments that were not included in the original DAMIP proposal circulated within the CMIP community (`http://wcrp-climate.org/images/modelling/WGCM/CMIP/ ApplicationSummary_CMIP6-EndorsedMIPs_150408_Sent.pdf`) i.e., tier 3 experiments - histCO2, histSOL and ssp245NAT. The inclusion of these experiments were also not discussed with other scientists at the IDAG (International Detection and Attribution Group) meeting held in February this year. The motivation for including these experiments should have wider community discussions, as it is not really clear how useful they are [See below specific comments about those experiments]. They may be done at the expense of more useful experiments.

More is needed to be said about the general type of analyses expected. In particular what is required for analyses using multi-model mean is not the same as what is needed for analyses on individual models. The ensemble size, especially for forcing factors with relatively weak response patterns, is much more important for analyses on individual models. As mentioned below, the recommendation of at least 3 ensemble members for histNAT, histSOL and histVOL etc. is not ideal for many types of analyses on individual models. However it may be more than sufficient for a multi-model mean analysis, where the total number of ensemble members will be much larger (e.g., Gillett et al, Journal of Climate [2013] and Jones et al, Journal of Geophysical Research [2013] both included both types of analysis using CMIP5 models).

**Specific comments**

Page 3 Line 10 and elsewhere. As explained in the manuscript, specific analysis plans are not given, but it would be helpful to indicate how the contributions to observed climate changes from well mixed greenhouse gases and also aerosols can be extracted with the proposed tier 1 experiments. It is mistakenly stated that previous studies separated contributions from WMGHGs and aerosols (Page 3 lines 12-13 and elsewhere). Those studies actually separated contributions from WMGHGS and other (non WMGHG) anthropogenic factors, as well as natural influences. It may seem pedantic, but in fact the contributions from ozone and land use changes are potentially substantial important for some diagnostics and should not be excluded if possible.

Page 3 Lines 15-17. I think the wrong Ribes paper has been referenced here in two places. Ribes and Terray, Clim. Dynam. [2013] is much more appropriate, as it has a detection study on observations and examines differences when using different models. Ribes et al. [2015] does neither of those things.

Page 3 lines 20-23. If ozone and land use are not included with aerosols, what should they be included with? Are the authors proposing ignoring these forcing factors?

Page 4 line 13 and elsewhere. The use of 'early 21st century slowdown' should be discouraged. Like other terms attempting to describe this period, e.g., 'hiatus', it is ill defined at best and inaccurate and misleading at worst. The scientific literature has tied itself up in knots about misleading names for this period. The authors should be really clear about what they mean.

Page 5 Lines 26-29 This is an excellent recommendation. It was generally difficult to deduce what models did follow after 2005 for the GHG experiments. Proposing a specific SSP will help to maintain consistency in the future simulations.

[Figure]

Page 6 Lines 1-5 Presumably (as is stated later) the natural radiative forcings for the 2015-2020 period will follow those as planned to be used by ScenarioMIP. It should be mentioned that future volcanic forcing is planned to be "ramped up" [O'Neill et al GMDD 2016] between 2015 and 2025 to control levels and that future solar irradiance will contain a repeat of past solar irradiance variations. It might actually make sense to push back the ramping up (or down) of the volcanic aerosol until after 2020, to avoid introducing an artificial signal into the models at the end of the historical period. Something to liaise with O'Neill et al [2016] about?

Page 6 Lines 12-13 Is it wise to put simulations covering the period 2015-2020 in CMIP6 with the label 'SSP2-4.5'? That is inconsistent with what is required by ScenarioMIP for that label, i.e., they expect simulations to cover the 2015-2100 period. This could cause confusion.

Page 6 lines 14-17 There may be some diagnostics or filtering combinations it will be ok, but for characterising multi-decadal near surface temperature patterns of change in individual models, 3 ensemble is likely to be insufficient, e.g., Ribes et al [2015].

Page 6 Lines 21-25 It is excellent to see this recommendation. It is also worth giving a recommendation for land use changes. For instance at least one CMIP5 model (IPSL-CM5A-LR) may have included historic land use change in their historicalGHG experiment.

Page 8 lines 10 onwards It is not clear how useful a small ensemble of histSOL will be for many detection studies. The authors are rather over confident when they say (line 24) that "unambiguous characterization of each model's solar signal" can be deduced and "separate clearly solar and volcanic effects". Having only three initial condition ensemble members will make this very difficult for many diagnostics and filtering choices (Ribes [2015]). If analysts are interested in impacts of just the solar cycle, then better experiments could be designed. The ssp245Nat for instance may provide the required data for those needs.

Page 8 lines 27 It is rather over confident to state "will allow the characterisation of and attribution to volcanic influences." Three initial condition ensemble members are likely to be insufficient for multi decadal near surface temperature analyses on individual models (Ribes [2015]).

Page 8 Line 32 onwards the usefulness of an ensemble of histCO2 is not at all obvious and I wonder if it should have been more widely discussed within the detection and modelling communities before being proposed. The authors main reason for inclusion of this experiment seems to be helping to constrain TCRE (Eq 2 in Gillett [2013]). The TCRE (temperature change relative to cumulative emission of $CO_2$) should be constant for a given model. So shouldn't the information provided by the 1% $CO_2$ experiment be sufficient? If delta T (1% run @ 2x)/E(2x) doesn't equal delta T (histCO2 @ 2010)/E(2010) then the usefulness of TCRE is itself questionable. Thus eqn 2 in Gillett [2013] should just be TCRE = beta * TCR / E(2x) The authors also give another reason - that it would help to better characterise the uncertainties in TCR. I am not sure this makes sense. histCO2 would help in understanding uncertainties in the predictive power of TCR estimating GHG warming, not the other way round.

Page 9 lines 9-22 Depending on whether analyses using ssp245AER and ssp245NAT are using multi-model means or individual models, just asking for one ensemble member (Table 1) may not be sufficient to characterizes responses accurately.

Page 9 lines 31-34 It should be mentioned that ozone and land use forcing factors will be folded up in the examples given, contributing to the "aerosol forcing uncertainty" and "natural forcing uncertainty".

Page 10 - section 3 Are there any MIPs that it would be vital to have involvement with? It is highlighted that RFMIP is close to DAMIP, but I wonder if it should be considered a vital MIP that should be done with the same model as used for DAMIP. There is a danger that some institutions will use one version of their model for some MIPs and other versions of their model for other MIPs.

**Technical comments**

Page 8 lines 12-13 The past solar cycle in solar irradiance has had cycle lengths of between 9 and 13 years.

Page 13 Line 9 Gregory et al should be on a new line.

line 14 page 15 Large font size

line 21 page 14 Submitted to what? I guess GMD?

line 32 page 15 Submitted to what?

———————————————————

---

## Short Comment (SC2) · 15 Jun 2016

The CMIP Panel is undertaking a review of the CMIP6 GMD special issue papers to ensure a level of consistency among the invited contributions, also in answering the key questions that were outlined in our request to submit a paper to all co-chairs of CMIP6-Endorsed MIPs. We very much welcome the important contribution from the DAMIP to CMIP6, and below are a few comments:

- p. 3 line 21: the D/A single forcing simulations were actually made available subsequent to CMIP3 in what was called CMIP4 (though not widely publicized as such). You can refer to Stouffer et al to fill in this chronology between CMIP3 and CMIP5: Stouffer, R.J., V. Eyring, G.A. Meehl, S. Bony, C. Senior, B. Stevens, K.E. Taylor, 2016: CMIP5 scientific gaps and recommendations for CMIP6. Bull. Amer. Meteorol. Soc., in press.

[Figure]

- p. 5 lines 1-10: the "All-but" approach could also include, for example, "all forcings but aerosols", and so on. It may be worth clarifying this distinction here. - P. 7 line 17: It may be worth clarifying here that natural forcings for the future would not include volcanoes unless I missed something, and only an estimate of solar; this comes up again on p. 9, lines 17-22 - P. 8 line 27: should be "HistVLC", not "HistVOL"

Updated Reference: - Eyring, V., Bony, S., Meehl, G. A., Senior, C. A., Stevens, B., Stouffer, R. J., and Taylor, K. E.: Overview of the Coupled Model Intercomparison Project Phase 6 (CMIP6) experimental design and organization, Geosci. Model Dev., 9, 1937-1958, doi:10.5194/gmd-9-1937-2016, 2016.

With many thanks for your ongoing efforts in the CMIP6 process.

The CMIP Panel

---

## Author Comment (AC1) · 16 Aug 2016

Executive Editor Comment: In agreement with the CMIP6 panel members, the Executive editors of GMD would like to establish a common naming convention for the titles of the CMIP6 experiment description papers. The title of CMIP6 papers should include both the acronym of the MIP, and CMIP6, so that it is clear this is a CMIP6-Endorsed MIP.

Additionally, we strongly recommend to add a version number to the MIP description. The reason for the version numbers is so that the MIP protocol can be updated later, normally in a second short paper outlining the changes.

Response: Title revised to: 'The Detection and Attribution Model Intercomparison Project (DAMIP v1.0) contribution to CMIP6'

---

## Author Comment (AC2) · 16 Aug 2016

We thank the reviewer for his helpful comments. We have revised the manuscript accounting for these comments.

**1. About event attribution. The paper provides a detailed enumeration of scientific questions for which DA experiments have been used in the past. Noticeably, the attribution of single weather / climate events has not been mentioned. I don't know if this was intentional or not. This area of event attribution has received much attention recently - maybe even excessively, I agree. What I would call the "dominant" method used to perform event attribution calculations involves large ensembles of forced atmospheric experiments - which are not DAMIP style simulations. But the counter-factual (ie NAT-only) SST are usually constructed using**

**a common DA analysis of long-term changes, in an ANT vs NAT decomposition. DAMIP like experiments are required here. Additionally, there have been efforts to assess how the final results (eg FARs) depend on the assumed (ANT and NAT) response patterns, which basically requires a MIP with histNAT simulations from different models. Overall, I think this is an important application, which requires DAMIP, and which could be mentioned.**

A link to the C20C+ Detection and attribution project was already mentioned in the manuscript. This link is now be described in more detail:

'This effort will be further facilitated by using output from DAMIP simulations as input to the C20C+ Detection and Attribution Project (Stone and Pall, 2016) and other similar projects, which use ensembles of simulations of atmosphere-only models driven using observed sea surface temperatures and sea ice, and other similar experiments in which attributable anthropogenic changes are removed from the prescribed SSTs and sea ice to quantify the contribution of anthropogenic changes to individual extreme events. Such studies rely on historical simulations and historical simulations with natural forcings only, such as those included in DAMIP.'

**2. Comprehensive list of external forcings and how they are clustered into subsets. The paper provides a very clear list of experiments, and in most cases, a clear list of forcing agents to be considered in these experiments. However, I suggest it might be useful to provide an "as exhaustive as possible" list of external forcings, and the individual forcing experiments in which they are supposed to be included (eg which are classed as "Aerosols", etc). I think this might be useful for several reasons: - there might be some inconsistencies with AR5, e.g., with respect to what is called aerosols. According to, e.g., the AR5 Fig 8.17, aerosols do not include NOx or NMVOC, while these species are included in the histAER experiment that is proposed here. The status of tropospheric Ozone with respect to GHGs was also somewhat unclear in CMIP5. -some external forcings are not included in any DAMIP individual forcing simulations. This**

**applies to LU, which is being treated in LUMIP, but also to Tropospheric Ozone, and maybe other short lived gases like CO. An exhaustive list could make this clearer. - NetCDF files from CMIP5 were usually specifying a list of forcing agents explicitly (at least for GHG species, in my memory), so this work would have to be done at some point anyway.**

A comprehensive list of forcings is now included in Table 1, as requested. The forcing agents to include in the HistAER simulations have now been clarified. NOx and NMVOCs have been included because they will affect the concentrations of nitrate aerosol and organic aerosol in models which resolve the relevant processes in their aerosol schemes. The description of histGHG has been revised to make even more clear that ozone is held fixed in these simulations: 'This will ensure that tropospheric and stratospheric ozone are held fixed in all these simulations, and simulated responses in models with and without coupled chemistry are comparable.'

**Other minor comments * p3 l15 and l17: I suspect that the appropriate reference is Ribes and Terray 2013, instead of Ribes et al. 2015.**

Thank you. These references have been replaced with references to Ribes and Terray (2013).

**\* p3 l21-22 "with those of ozone and land use changes": I suggest replacing by "with the response to other external forcings, most notably ozone and land use" or something of that effect, as I'm not sure that all forcings can be put into the categories GHG, AER, NAT, OZ and LU (see also comment 2).**

Suggested change made.

**\* p4 l2: "the other" is written twice**

One instance of 'the other' deleted.

**\* p5 1st paragraph: I suggest adding somewhere something like "The two approaches ["only" vs "all-but"] are equivalent if additivity holds, but might differ**

**otherwise."**

Suggested phrase inserted, but with the additional proviso that the equivalence only holds in the limit of large ensembles. With small ensembles the noise in the response pattern will depend on whether it is simulated directly, or is calculated from a residual. Inserted phrase: 'The two approaches yield equivalent results if additivity holds and in the limit of large ensembles, but can differ otherwise.'

**\* p5 Sentence l5-7: it is not clear to me that the "all-but" approach is more appropriate than the "only" one for the second question mentioned... Is it what the authors want to say? As an alternative, causality theory wight be mentioned explicitly, as it seems to be the main motivation leaning towards an "all-but" approach.**

The latter part of the phrase has been revised to read: 'or to detect the contribution of a particular forcing to observed climate change.' In this case the "all-but" formulation is clearly preferable, since the assumption of additivity is not required. We consider that a discussion on causality theory would be beyond the scope of this manuscript.

**\* p5 l14: "linear additivity" has been indeed commonly discussed in the literature but it seems to me that, strictly speaking, only "additivity" is assumed in the experimental design. "Linearity", in my view, is more related to the use of analysis techniques based on linear regression, like optimal fingerprinting.**

'linear additivity' replaced with 'additivity'.

**\* p5 l21-22: Does this also apply to GHGs other than CO2?**

Text amended to clarify this point: 'All simulations used in DAMIP are driven by $CO_2$ concentration rather than $CO_2$ emissions. In common with all other CMIP6 simulations, concentrations of the other well-mixed greenhouse gases (WMGHGs) are also specified in DAMIP simulations.'

**\* p6 histALL: This is probably well specified in other CMIP6 documents, but I**

**think it might be useful to add a quick description of the NAT forcing recommended in SSP. To my knowledge, there were no clear recommendations in CMIP5 regarding the volcanic forcing. Additionally, at least one modelling center decided to run historicalExt experiments with no volcanoes (consistent with the observation of no major eruption when the run was realised in 2011/12), while the RCPs were run with a volcanic background. This led to historicalExt runs which differ from the corresponding RCP runs over their common period. My understanding of the description given here is that such a discrepancy should be avoided in CMIP6, and I think it would be useful to state this even more clearly.**

The natural forcings to be used in the 2015-2020 period have now been clarified: 'Time-evolving solar forcing, and stratospheric aerosol ramping up towards the piControl background level should be prescribed over the 2015-2020 period as prescribed by ScenarioMIP (O'Neill et al., 2016).', and a reference to O'Neill et al. (2016) has been added to the references list.

**\* p6 histGHG: To make the point even clearer, I suggest adding a sentence such as "Ozone (tropospheric and stratospheric) is excluded from GHG species [and is therefore supposed to remain roughly constant in these experiments]", eg at l21.**

The description already had a sentence noting that ozone is held fixed in these experiments. This point has now been made even more clearly by amending this sentence to read: 'This will ensure that tropospheric and stratospheric ozone are held fixed in all these simulations'.

**\* p7 l17: maybe add "(see Tier 3)" after ssp245NAT**

'(see Section 2.3)' inserted to refer the reader to the description of the Tier 3 experiments.

**\* Tiers 2 and 3: information on minimum ensemble sizes seems less precise for**

**those Tiers if compared to Tier 1 - maybe it might be said somewhere that the general rule is at least 3 for historical, at least (only) 1 for SSP... Or maybe I missed it?**

In the introductory paragraphs of sections 2.2 and 2.3 we have inserted 'Minimum ensemble sizes are three for the historical simulations and one for the future simulations.' Note that this information was already included in Table 1, but we have now included it in the text for completeness.

**\* Lastly, I also suggest adding somewhere something like "Tropospheric ozone (and possibly other external forcings, if any) is not considered in any of the historical experiments driven by subsets of forcings which are proposed in DAMIP.". Consequently, quantifying the response to this forcing would require subtraction, with a possible confounding effect related to non-additive responses to other forcings. Note that this choice doesn't seem inappropriate to me, but just it would be useful to make it clear in order to prevent misinterpretation.**

We have added the following sentence to the description of histSOZ: 'Note that DAMIP does not include simulations isolating the effects of tropospheric ozone changes.'

---

## Author Comment (AC3) · 16 Aug 2016

We thank the reviewer for his or her helpful comments. We have revised the manuscript accounting for these comments.

**General comments.    I fear, with respect to resources required, that the proposal is somewhat over ambitious.  Just for Tier 1 the number of experiments and initial condition ensembles required, >1870 model years, may substantially limit the number of models participating in DAMIP. ScenarioMIP (O'Neill GMDD [2016]) say "the success of ScenarioMIP lies in the broad participation of the CMIP6 modelling groups in Tier 1 experiments...". Should the DAMIP plan have the same ambition?  A much smaller population of experiments/ensembles for a 'Tier 0', to focus on a few important scientific questions, could encourage as**

**wide a range of models to take part in DAMIP as possible. The remaining tiers can then be populated by institutions with more resources. It would be a shame to miss the opportunity to design experiments that would encourage greater institution involvement than there was for CMIP5 detection analyses. A 'lessons learned' exercise and finding out why some institutions didn't produce 'detection' experiments for CMIP5 might have been helpful.**

We respect the reviewer's opinion, but in our view including at least the three Tier 1 experiments described is a minimum requirement for participation in DAMIP. Ensembles of size one are of limited use for attribution, and we really feel that histNAT, histGHG and histAER are required at Tier 1. It is more important to have a complete set of simulations from a slightly more limited set of models, than to have small ensembles and individual experiments from a broader range of models. Initial consultations with modelling groups indicated anticipated participation from at least as many modelling groups as carried out equivalent simulations in CMIP5.

**There are several experiments that were not included in the original DAMIP proposal circulated within the CMIP community i.e., tier 3 experiments - histCO2, histSOL and ssp245NAT. The inclusion of these experiments were also not discussed with other scientists at the IDAG (International Detection and Attribution Group) meeting held in February this year. The motivation for including these experiments should have wider community discussions, as it is not really clear how useful they are [See below specific comments about those experiments]. They may be done at the expense of more useful experiments.**

The document referred to above was a proposal, and the CMIP6 panel rules indicated that additional experiments could be added at Tiers 2 or 3 prior to preparation of the MIP description paper. Note that the experiments described above were all added at the lowest priority Tier 3 level, so whether or not to carry out these experiments is at the discretion of the modelling groups, and their presence is unlikely to discourage a modelling group from taking part in DAMIP. Taking each of the additional experiments referred to in turn: histSOL was added to DAMIP after SolMIP was merged with DAMIP by the CMIP6 panel. ssp245NAT was added to the experimental design when it became clear that ScenarioMIP simulations would include time-varying solar and volcanic forcings (O'Neill et al., 2016). We had previously assumed that natural forcings would be held constant in the SSP simulations. These simulations are required in order to separate the effects of projected changes in anthropogenic emissions on future climate from the effects of projected changes in natural forcings. They are also required for observationally-constrained projections using the Allen, Stott and Kettleborough approach in which natural forcings are treated separately. histCO2 simulations were added after consultation among the authors in order to improve constraints on TCR and TCRE as described. HistSOL was presented at IDAG, though the referee is correct that the other two experiments were added after the IDAG meeting.

**More is needed to be said about the general type of analyses expected. In particular what is required for analyses using multi-model mean is not the same as what is needed for analyses on individual models. The ensemble size, especially for forcing factors with relatively weak response patterns, is much more important for analyses on individual models. As mentioned below, the recommendation of at least 3 ensemble members for histNAT, histSOL and histVOL etc. is not ideal for many types of analyses on individual models. However it may be more than sufficient for a multi-model mean analysis, where the total number of ensemble members will be much larger (e.g., Gillett et al, Journal of Climate [2013] and Jones et al, Journal of Geophysical Research [2013] both included both types of analysis using CMIP5 models).**

As described at the end of Section 1, we do not include a complete analysis plan because the detection and attribution community is already well-established, and its activities are already coordinated through IDAG, and to address the long-standing interest in attribution in the IPCC reports. Nontheless, anticipated analyses are described in paragraphs 4-6 of Section 1. Additional details on the usage of the histAER simula-
tions, and the usage of DAMIP simulations for event attribution have now been added to these paragraphs. We anticipate that DAMIP simulations will be used both for individual model attribution studies and multi-model attribution studies. The referee is correct that large ensemble sizes are more important for single model analyses than those based on multiple models. Nonetheless, approaches which attempt to account for model uncertainty would benefit from the availability of initial condition ensembles in order to more robustly separate uncertainties in the response patterns associated with internal variability from those associated with model uncertainty (e.g. Hannart et al., Geophys. Res. Lett., 2014). We thus prefer to retain a minimum ensemble size of three for the historical simulations.

**Specific comments Page 3 Line 10 and elsewhere. As explained in the manuscript, specific analysis plans are not given, but it would be helpful to indicate how the contributions to observed climate changes from well mixed greenhouse gases and also aerosols can be extracted with the proposed tier 1 experiments. It is mistakenly stated that previous studies separated contributions from WMGHGs and aerosols (Page 3 lines 12-13 and elsewhere). Those studies actually separated contributions from WMGHGS and other (non WMGHG) anthropogenic factors, as well as natural influences. It may seem pedantic, but in fact the contributions from ozone and land use changes are potentially substantial important for some diagnostics and should not be excluded if possible.**

To address the reviewer's comment regarding how the Tier 1 simulations may be used, the following text has been added on page 3: 'Such aerosol-only simulations may be used together with historical simulations including all forcings and historical simulations with natural forcings only to estimate attributable contributions to observed changes due to natural forcings, due to aerosols, and due to the combined effects of well-mixed greenhouse gases, ozone and land-use changes. Since some part of the greenhouse gas changes is associated with land-use change, and since ozone is a greenhouse gas, grouping these forcing together arguably makes more sense than grouping ozone

and land-use change with aerosols.' As stated in the description of histGHG, 'histALL, histNAT and histGHG will allow the attribution of observed climate change to natural, greenhouse gas and other anthropogenic components.' Note that using all four Tier 1 experiments described here it would also be possible to carry out an attribution analysis of observed changes to natural forcings, to WMGHGs, to aerosols, and to ozone and land-use change combined. However, we expect that regression coefficients would likely be unconstrained in such a four-pattern regression, therefore we do not propose it here.

The text referred to on page 3 has been revised to read: 'While some earlier studies were able to clearly separate the influences of greenhouse gases and other anthropogenic forcings on observed temperature changes using individual models (Stott et al., 2006), more recent studies using newer models and a longer period of observations have identified substantial uncertainties in the separate estimation of greenhouse gas and other anthropogenic contributions, where the other anthropogenic contribution is dominated by aerosols but also includes the response to ozone changes and land use changes in most models (Jones et al., 2013; Gillett et al., 2013; Ribes and Terray, 2015)'.

**Page 3 Lines 15-17. I think the wrong Ribes paper has been referenced here in two places. Ribes and Terray, Clim. Dynam. [2013] is much more appropriate, as it has a detection study on observations and examines differences when using different models. Ribes et al. [2015] does neither of those things.**

The reference has been replaced as recommended.

**Page 3 lines 20-23. If ozone and land use are not included with aerosols, what should they be included with? Are the authors proposing ignoring these forcing factors?**

In this approach ozone and land use change are grouped with WMGHGs. This comment has been addressed by the insertion of the following two sentences: 'Such

aerosol-only simulations may be used together with historical simulations including all forcings and historical simulations with natural forcings only to estimate attributable contributions to observed changes due to natural forcings, due to aerosols, and due to the combined effects of well-mixed greenhouse gases, ozone and land-use changes. Since some part of the greenhouse gas changes is associated with land-use change, and since ozone is a greenhouse gas, grouping these forcing together arguably makes more sense than grouping ozone and land-use change with aerosols.'

**Page 4 line 13 and elsewhere. The use of 'early 21st century slowdown' should be discouraged. Like other terms attempting to describe this period, e.g., 'hiatus', it is ill defined at best and inaccurate and misleading at worst. The scientific literature has tied itself up in knots about misleading names for this period. The authors should be really clear about what they mean.**

Two instances of 'slowdown period' replaced by 'early 21st century' and one instance of 'early 21st–century slowdown of climate warming' replaced by 'period of reduced warming in the early 21st–century' to address the reviewer's comment.

**Page 5 Lines 26-29 This is an excellent recommendation. It was generally difficult to deduce what models did follow after 2005 for the GHG experiments. Proposing a specific SSP will help to maintain consistency in the future simulations.**

Noted – we are pleased the reviewer supports this recommendation.

**Page 6 Lines 1-5 Presumably (as is stated later) the natural radiative forcings for the 2015-2020 period will follow those as planned to be used by ScenarioMIP. It should be mentioned that future volcanic forcing is planned to be "ramped up" [O'Neill et al GMDD 2016] between 2015 and 2025 to control levels and that future solar irradiance will contain a repeat of past solar irradiance variations. It might actually make sense to push back the ramping up (or down) of the volcanic aerosol until after 2020, to avoid introducing an artificial signal into the models at the end of the historical period. Something to liaise with O'Neill et al [2016]**

**about?**

A description of the evolution of natural forcings over the 2015-2020 period in histALL has now been added to the description of histALL 'Time-evolving solar forcing, and stratospheric aerosol ramping up towards the piControl background level should be prescribed over the 2015-2020 period as specified by ScenarioMIP (O'Neill et al., 2016).', including a reference to O'Neill et al. (2016). We have requested that ScenarioMIP move the ramp up of aerosols to after 2020 as suggested, but do not yet know whether this change will be made.

**Page 6 Lines 12-13 Is it wise to put simulations covering the period 2015-2020 in CMIP6 with the label 'SSP2-4.5'? That is inconsistent with what is required by ScenarioMIP for that label, i.e., they expect simulations to cover the 2015-2100 period. This could cause confusion.**

For groups who are running an ensemble of SSP2-4.5 simulations of the same size as the ensemble of historical simulations this will not be a problem. We have asked ScenarioMIP whether they are concerned about this in the case that some modelling groups participate in DAMIP, but do not run the same size ensemble of SSP2-4.5 simulations. They have indicated that this is not a concern.

**Page 6 lines 14-17 There may be some diagnostics or filtering combinations it will be ok, but for characterising multi-decadal near surface temperature patterns of change in individual models, 3 ensemble is likely to be insufficient, e.g., Ribes et al [2015].**

We agree with the reviewer that an ensemble size for historicalNAT of greater than three is desirable, but as the reviewer states elsewhere, setting the minimum ensemble size too high could discourage participation from modelling groups. Clearly we need to set a balance. It is expected that some modelling groups will run ensembles larger than the minimum requested size of three.

[Figure]

**Page 6 Lines 21-25 It is excellent to see this recommendation. It is also worth giving a recommendation for land use changes. For instance at least one CMIP5 model (IPSL-CM5A-LR) may have included historic land use change in their historicalGHG experiment.**

Thanks for the positive comment. As requested by reviewer 1, we now include a full list of forcings to be perturbed in each experiment in Table 1, clearly indicating that land use change is changed in histALL, but not in histGHG.

**Page 8 lines 10 onwards It is not clear how useful a small ensemble of histSOL will be for many detection studies. The authors are rather over confident when they say (line 24) that "unambiguous characterization of each model's solar signal" can be deduced and "separate clearly solar and volcanic effects". Having only three initial condition ensemble members will make this very difficult for many diagnostics and filtering choices (Ribes [2015]). If analysts are interested in impacts of just the solar cycle, then better experiments could be designed. The ssp245Nat for instance may provide the required data for those needs.**

This experiment was adopted from SolarMIP. As noted in the text, if analysts are interested in identifying the historical simulated response to solar variations, then simulations with solar forcing only are required. ssp245Nat could be used to identify a model's response to solar forcing variations, but this would be for projected future variations, rather than observed historical variations. As noted previously three is a minimum ensemble size, and modelling groups interested in separating the solar signal could run a larger ensemble. Nonetheless, in response to the reviewer's comment, 'umambiguous characterisation' and 'separate clearly' were moderated so that the text now reads 'The proposed histSOL experiment will facilitate the characterization of each model's solar signal and allow the separation of solar and volcanic effects over the historical period.'

**Page 8 lines 27 It is rather over confident to state "will allow the characterisation of and attribution to volcanic influences." Three initial condition ensemble**

**members are likely to be insufficient for multi decadal near surface temperature analyses on individual models (Ribes [2015]).**

The reviewer is correct – we meant to indicate that the simulations would allow detection and attribution studies on volcanic influence, rather than guaranteeing that volcanic influence will be detectable. Revised to read 'The histVLC experiments will facilitate detection and attribution studies on volcanic influence.'

**Page 8 Line 32 onwards the usefulness of an ensemble of histCO2 is not at all obvious and I wonder if it should have been more widely discussed within the detection and modelling communities before being proposed. The authors main reason for inclusion of this experiment seems to be helping to constrain TCRE (Eq 2 in Gillett [2013]). The TCRE (temperature change relative to cumulative emission of CO2) should be constant for a given model. So shouldn't the information provided by the 1experiment be sufficient? If delta T (1@ 2010)/E(2010) then the usefulness of TCRE is itself questionable. Thus eqn 2 in Gillett [2013] should just be TCRE = beta \* TCR / E(2x) The authors also give another reason - that it would help to better characterise the uncertainties in TCR. I am not sure this makes sense. histCO2 would help in understanding uncertainties in the predictive power of TCR estimating GHG warming, not the other way round.**

The reviewer misunderstands equation 2 in Gillett et al. (2013). One way of estimating TCRE from observations is to calculate the ratio of CO2-attributable warming to date, to an estimate of cumulative CO2 emissions to date. This calculation relies on TCRE being approximately constant as a function of cumulative emissions. In order to calculate CO2-attributable warming, simulations of the response to historical changes in CO2 only are required. The only other way to calculate CO2-attributable warming would be to multiply GHG-attributable warming by the ratio of CO2 to total GHG forcing (as in Matthews et al., Nature, 2009), but this calculation assumes that the efficacy of all GHGs is one, and it assumes that the temperature response to CO2 and the other GHGs is proportional to their present-day forcing, which is not generally true, given

that the time evolution of their radiative forcings has been different. While we could use simulations from a single model to estimate the ratio of CO2-attributable to GHG-attributable warming (as in Gillett et al., 2013), efficacies differ amongst models, as does the climate response, and thus a much more robust approach would be to use a multi-model ensemble to assess this ratio and its uncertainty.

The reviewer asks why eq (2) in Gillett et al. (2013) cannot be replaced with TCRE = beta * TCR / E(2x). Presumably in this equation the reviewer means that TCR is assessed from one or more models and E(2x) is assessed from the same models, while beta is the regression coefficient obtained from a detection and attribution analysis applied to global temperature. This would be one way to assess TCRE from observations, but it would only use temperature information from observations, and not use any observations of the carbon cycle. Equation 2 from Gillett et al. (2013) uses an estimate of actual emissions from observations (rather than diagnosed emissions from a model), and hence uses observations of both carbon cycle and physical climate to constrain TCRE.

Regarding TCR, approaches to estimating TCR from observations typically calculate a scaling factor on well-mixed greenhouse gases from a regression of observed temperature change onto the simulated response to WMGHGs, as well as other anthropogenic forcings and natural forcings. They then use this regression coefficient to scale a model estimate of TCR (i.e. the simulated response to 1PCTCO2 at doubled CO2), without accounting for the additional uncertainty introduced by this step. This approach would be valid if there were a perfect relationship between historical warming due to WMGHGs and TCR across models. But there is a pronounced spread in this ratio (Gillett et al., 2013; Figure 6). It is not clear whether the spread in this ratio comes mainly from differences in radiative forcings or efficacies between different GHGs across models, or from differences in the temporal response to different time evolution of the radiative forcings of the different GHGs. A multi-model ensemble of CO2-only simulations would allow us to address this issue. The reviewer notes such

simulations could also be used to understand the relationship between TCR and pre-
dicted GHG-induced warming – this is correct – but the simulations would be just as
useful for understanding the sources of uncertainty in observationally-constrained es-
timates of TCR.

Hence, CO2-only simulations would help us characterize and understand the uncer-
tainties in observationally-constrained estimates of TCRE and TCR, both of which are
important policy-relevant metrics, which were both extensively discussed in the Sum-
mary for Policymakers of the Working Group I contribution to the IPCC Fifth Assess-
ment Report, for example.

Finally regarding the question of consultation – these simulations were added after the
initial proposal was prepared, but were added with consultation of the DAMIP commit-
tee. Note that they are included only at the lowest Tier 3 priority, and therefore can be
carried out at the discretion of modelling groups and are unlikely to dissuade modelling
groups from carrying out DAMIP.

**Page 9 lines 9-22 Depending on whether analyses using ssp245AER and
ssp245NAT are using multi-model means or individual models, just asking for
one ensemble member (Table 1) may not be sufficient to characterizes responses
accurately.**

Clearly there is a balance to be struck between asking for large ensembles to better
account for internal variability, and asking for small ensembles to encourage the par-
ticipation of the largest number of modelling groups. In an observationally-constrained
projection exercise, large ensembles are less critical for the future simulations than for
the historical portion of the simulations where the ensemble size may strongly influence
the uncertainties on the regression coefficients on individual forcings. For this reason
we chose to only request a minimum ensemble size of one for the future simulations.
Note that this is a minimum ensemble size, and modelling groups are free to carry out
larger ensembles.

We have added this final sentence to the introductory paragraph of Section 2.2 to make clear that larger ensembles are encouraged: 'Minimum ensemble sizes are three for the historical simulations and one for the future simulations, though modelling groups are encouraged to run larger ensembles if resources allow.'

**Page 9 lines 31-34 It should be mentioned that ozone and land use forcing factors will be folded up in the examples given, contributing to the "aerosol forcing uncertainty" and "natural forcing uncertainty".**

We are not proposing that alternative ozone or land use forcings are used in either histALLestAER2 or histALLestNAT2. If alternate estimate of these forcings were available it would be possible to investigate the role of these forcings in introducing uncertainty in attribution results. However, after consultation with the community, including at the IDAG meeting, we assess that uncertainties in aerosols and natural forcings have made the largest contribution to uncertainties in global temperature changes, and it was necessary to focus on a limited number of experiments here. Thus ozone and land use forcings are not contributors to the estimates of 'aerosol forcing uncertainty' and 'natural forcing uncertainty' that may be derived using these experiments.

**Page 10 - section 3 Are there any MIPs that it would be vital to have involvement with? It is highlighted that RFMIP is close to DAMIP, but I wonder if it should be considered a vital MIP that should be done with the same model as used for DAMIP. There is a danger that some institutions will use one version of their model for some MIPs and other versions of their model for other MIPs.**

Other than the DECK and CMIP6 historical simulation there are no other simulations that it is vital that participating groups carry out. ScenarioMIP is the most strongly-linked MIP, since it provides the SSP simulations which are needed for observationally-constrained projections carried out using some of our Tier 2 and 3 simulations. But given that these simulations are optional in DAMIP, participation in ScenarioMIP is not required. Participation of models in DAMIP and in the other MIPs included in Table 2

(including RFMIP) would have benefits as described there, but participation in these other MIPs is not vital.

**Technical comments Page 8 lines 12-13 The past solar cycle in solar irradiance has had cycle lengths of between 9 and 13 years.**

'11-year cycle' replaced with 'approximately 11-year cycle'.

**Page 13 Line 9 Gregory et al should be on a new line.**

Thanks for pointing this out. Corrected.

**line 14 page 15 Large font size**

Thanks for pointing this out. Corrected.

**line 21 page 14 Submitted to what? I guess GMD?**

The journal was already specified – GMD. The reference has now been updated with a doi.

**line 32 page 15 Submitted to what?**

The journal the paper was originally submitted to did not publish it since it was judged out of scope. The paper is currently in preparation for submission to another journal, and the reference has been revised accordingly.
* * *

---

## Author Comment (AC4) · 16 Aug 2016

Thanks to the CMIP6 Panel for their comments. We have revised the manuscript taking into account their suggestions.

**- p. 3 line 21: the D/A single forcing simulations were actually made available subsequent to CMIP3 in what was called CMIP4 (though not widely publicized as such). You can refer to Stouffer et al to fill in this chronology between CMIP3 and CMIP5: Stouffer, R.J., V. Eyring, G.A. Meehl, S. Bony, C. Senior, B. Stevens, K.E. Taylor, 2016: CMIP5 scientific gaps and recommendations for CMIP6. Bull. Amer. Meteorol. Soc., in press.**

A reference to Stouffer et al. has been inserted where the DA single forcing simulations following CMIP3 are discussed.

[Figure]

**- p. 5 lines 1-10: the "All-but" approach could also include, for example, "all forcings but aerosols", and so on. It may be worth clarifying this distinction here.**

We have inserted the following text to provide an example of the 'all-but' approach: 'An example of the latter is the LUMIP land-NoLu simulation which includes changes in all forcings but land use change (Lawrence et al., 2016).'

**- P. 7 line 17: It may be worth clarifying here that natural forcings for the future would not include volcanoes unless I missed something, and only an estimate of solar; this comes up again on p. 9, lines 17-22**

According to O'Neill et al. (2016): 'Volcanic forcing will be ramped up from the value at the end of the historical simulation period (2015) over 10 years to the same constant value prescribed for the piControl simulations in the DECK, and then will be kept fixed', hence volcanic aerosol does change in the SSP2-4.5 and ssp245NAT simulations up until 2025. This is now clarified in the description of ssp245NAT: 'The future solar forcing data recommended for CMIP6 has a downward trend (Matthes et al., 2016), and stratospheric aerosol is prescribed to ramp up from its level in 2014 to the background level specified in piControl over the 2015-2025 period (O'Neill et al., 2016).'

**- P. 8 line 27: should be "HistVLC", not "HistVOL"**

Thanks. This has been corrected.

**Updated Reference: - Eyring, V., Bony, S., Meehl, G. A., Senior, C. A., Stevens, B., Stouffer, R. J., and Taylor, K. E.: Overview of the Coupled Model Intercomparison Project Phase 6 (CMIP6) experimental design and organization, Geosci. Model Dev., 9, 1937-1958, doi:10.5194/gmd-9-1937-2016, 2016.**

Thanks. The reference has been updated.

---

## Author Response (AR1)

Responses to reviewer comments, including discussion of changes made to the manuscript:
**AC1**: 'Response to Executive Editor Comment', Nathan P. Gillett, 16 Aug 2016
**AC2**: 'Response to Aurelien Ribes', Nathan P. Gillett, 16 Aug 2016
**AC3**: 'Response to Referee #2', Nathan P. Gillett, 16 Aug 2016
5   **AC4**: 'Response to CMIP6 Panel', Nathan P. Gillett, 16 Aug 2016

Some additional changes were made to the manuscript beyond those described in the interactive discussion in order to more fully address the review comments, and based on further feedback and advice from the CMIP Panel and coauthors.
Additional changes made to the manuscript:
10  - References were carefully checked and updated or corrected where required. References to other MIP description papers in GMD were added.
- Experiment names were revised as requested by CMIP Panel Member Karl Taylor, in order to be consistent with those which will be used in CMIP6 file names. Figure 1 was revised accordingly.
- Rationale for extending historical simulations to 2020 explained in Section 2.1: 'A finish date of 2020 was chosen because
15  it will allow contemporary observations to be included in detection and attribution studies cited in the Sixth IPCC Assessment Report, without extending too far into the future.'
- Description of "only" and "all-but" approaches, in the opening of Section 2 further edited to improve clarity (see tracked changes below).
- A sentence on how different ensemble members should be initialised was added to the beginning of Section 2.1: 'We
20  recommend that ensemble members are initiated by choosing well-separated initial states from a control simulation.'
- Additional text added to the description of hist-sol to better characterise uncertainties 'and there is uncertainty in the significance and amplitude of the climate response to a solar changes due to the presence of internal variability' and to describe the advantage of hist-sol compared to hist-nat 'The hist-sol simulations will also allow the estimation of the effect of solar forcing on observations separately from that of volcanism (rather than in combination with volcanism using hist-
25  nat), which is essential for the quantification of the solar signal over the historical period.'
- Table 2 was updated to reflect newly available information on the other MIPs, based on their published GMD papers.
- Other minor changes were made to the text to improve clarity and readability.

A version of the manuscript with changes tracked follows.
30

[revised manuscript text omitted]
 histALL historical and histNAThistorical natural-only experiments, an "all-but" design. HoweverWhereas, the response to greenhouse gas forcing

[revised manuscript text omitted]
 aAllow observationally-constrainedts of uncertainties in future projections to be derived. |
| AerChemMIP[3] | HISTghg, HISTghgNtcf, WMFORCch4, WMFORCn2ohist-piNTCF hist-piAer hist-1950HC | historicalALL, histGHG, histSOZhist-nat | HISTghg hist-piNTCF (AerChemMIP) is the same as histALL historical (DAMIP) but with 1850 aerosol and tropospheric ozone precursors. In HISTghgNtcfhist-piAer, only the aerosol precursors are kept at 1850, while the ozone precursors follow the historical. WMFORCch4 and WMFORCn2o hist-1950HC is the same as histALL historical but with 1950 CFC and HCFC concentrations.experiments should also include the chemical effects on strat+trop ozone and strat H2O. These AerChemMIP simulations may be used with DAMIP simulations to attribute observed changes to changes in emissions of aerosol precursors, ozone precursors, or CFC and HCFCs, in combination with natural forcings and other anthropogenic forcings. |
| DCPP[4] | Historical+SSP2-4.5 C1.9 (Pacemaker Pacific experiment) C1.10 (Pacemaker Atlantic experiment) | historicalALL | DCPP proposes a 10-member ensemble of histALL historical up to 2030 also extended with SSP2-4.5. The combinations of DAMIP and DCPP/GMMIP experiments allow the assessments of the relative contributions of external forcing factors and the the response to the PDO and AMO to internal variability on the historical climate change. |
| GMMIP[5] | AMIP20C, HIST-IPO, HIST-AMO | historicalALL, hist-natNAT, hist-GHG, hist-aerAER | The combinations of DAMIP and DCPP/GMMIP allow the assessments of the relative contributions of external forcing factors and the internal variability toon the historical climate change. |
| LUMIP[6] | hist-NoLuLND_noLULCC-hist | historicalALL | "ALL minus land-use (hist-NoLuLND_noLULCC-hist)" of LUMIP and histALL historical will allow the separation of the effects of land-use changes and the response to other forcingsothers. |
| RFMIP-ERF[7] | RFMIP-ERF-HistAll, RFMIP-ERF-HistNat, RFMIP-ERF-HistGHG, RFMIP-ERF-HistAerHistorical+SSP2-4.5, Natural, Aerosols, WMGHG | historicalALL, histNAT-nat, hist-GHG, histAER-aer, ssp245-GHG, ssp245-aerAER, ssp245-natNAT | Combining radiative forcing estimated from RFMIP-ERF and transient climate responses from DAMIP, we can investigate how feedbacks and adjustments vary with forcing factors. |
| RFMIP-SpAer[7,8]Historical | RFMIP-SpAerO3-all, RFMIP-SpAerO3-aerHistorical, Hist-Nat, Hist-Aer | historicalALL, hist-natNAT, hist-aerAER | Combinations of DAMIP and RFMIP-Historical SpAer will allow us to separate uncertainties in climate response based on specified aerosol evolution from the overall uncertainties in climate response to specified aerosol precursor emissions. |
| GeoMIP[9] & VolMIP[10] | All | hist-volcVLC | The volcanic response of models can be validated against observations using histVLC, whereas GeoMIP experiments cannot. Thus histVLC experiments will provide useful context for interpreting simulated responses to stratospheric aerosol across models in the GeoMIP experiment. While VolMIP includes simulations of individual eruptions, it does not include simulations of the transient response to historical eruptions and its focus is on 19th century eruptions. histVLC facilitates validation of long-term transient effects against observations. |

---

## Author Response (AR2)

**Response to topical editor**

Topical editor's requested revision made (deletion of a single instance of 'a').

The manuscript was proofread again, and a small number of additional grammatical and spelling errors were corrected.